# Validation of the mental health continuum-short form: The bifactor model of emotional, social, and psychological well-being

**Zack Zhishen Yeo**[ID]*[◎], **Lidia Suárez**[ID][◎]

Department of Psychology, School of Social and Health Sciences, James Cook University, Singapore, Singapore

◎ These authors contributed equally to this work.

* zackyeozhishen@gmail.com

## Abstract

The Mental Health Continuum-Short Form (MHC-SF) is aimed at measuring the three dimensions of mental health; emotional, social, and psychological well-being. The purpose of the current study was to evaluate the psychometric properties of the MHC-SF within the context of Singapore and Australia. A total of 299 Singaporeans or permanent residents (59.2% female; mean age = 24.26, $SD$ = 6.13) and 258 Australians or permanent residents (69% female; mean age = 23.95, $SD$ = 8.66) completed the study. Confirmatory factor analyses were used to assess the structural validity of the MHC-SF. Internal consistency reliability was assessed via the Cronbach's α and MacDonald's ω reliability coefficients. Concurrent validity was examined against the World Health Organisation-Five Well-Being Index, discriminant validity using the Hospital Anxiety and Depression Scale, and criterion validity using a self-rated question of "Please rate your averaged level of mental health over the past month", all via Pearson's correlations. A bifactor model of the MHC-SF, where each item loaded on a general factor and simultaneously on their respective uncorrelated group factors, yielded the best fit to the data across both samples. Further investigations demonstrated that the general well-being factor accounted for majority of variances of the MHC-SF. Internal consistency reliability, concurrent validity, discriminant validity, and criterion validity were all demonstrated. In conclusion, the current study provided support for the bifactor model of MHC-SF and demonstrated evidence of good psychometrics across both samples. The results highlighted the unidimensionality of the measure, suggesting that it is more informative to interpret the aggregated score than scores of independent factors standalone.

## Introduction

Prevalence of mental disorders have significantly increased over the years [1, 2] and have seen an increase of attention with regard to research, policies, diagnosis, and treatments, while investigations into mental health have been side-lined [3–5]. The oversight has negatively

**Data Availability Statement:** The data of the study have been made public and can be found on figshare's depository. https://doi.org/10.6084/m9.figshare.19174205.v2.

**Funding:** We would like to update the funding received for the study. The publication fee has been sponsored by James Cook University (Singapore), School of Social and Health Sciences. The authors (Zhishen Yeo and Lidia Suarez) have been granted 2365 Singapore Dollars for the publication of the paper. The grant is called Publication Support Application and the application number is PSA20220002. The URL to the funder's website is https://www.jcu.edu.sg/. The funders had no role in study design, data collection and analysis, decision to publish, or preparation of the manuscript.

**Competing interests:** The authors have declared that no competing interests exist.

impeded our overall ability to address the increasing deterioration of mental health amongst all ages [6–9]. Disruptions to mental health are multi-layered and present differently to the varying age groups. Inadequacies in nurturing and attachment are major risk factors to negative well-being in children [6], while adolescents are faced with challenges such as academic pressure, and exposure to risky behaviours [10, 11]. Young adults encounter occupational, financial, and social difficulties [12–14], while older adults experience the deterioration of physical health, and perceived isolation [15, 16].

Overall, challenges to maintain a positive well-being are encountered throughout the lifespan, and considering the relationship of mental health and mental disorders, a valid and reliable standardised measure of well-being is required.

## Mental health and well-being

Mental health has traditionally been conceptualised as the absence of psychopathology [17, 18]. However, it has changed in recent years; the two-continua model of mental health and mental illness states that mental health is related to, but different from, mental illness: where one continuum represents the individual's state of mental health, while the other indicates the absence or presence of mental illness [19].

Keyes [19] investigated the relationship between mental health and mental illness using the representative data of 3,032 American adults between the ages of 25 and 74, obtained from the Midlife Development in the United States (MIDUS) survey. Confirmatory factor analyses demonstrated two factors correlating negatively, where measures of well-being loaded on one factor, while measures of psychopathology loaded on a second factor. This result is expected as mental health and illness while distinct, are strongly related, and mental health status have shown to be predictive of mental illness [20–22]. Doré, O'Loughlin [17] concluded that moderate to high inverse correlations between measures of mental health and mental illness to be supportive of the two-continua model and indicative of the divergence between the two constructs.

The structural results of the two-factor model of mental health and mental illness have been replicated over different samples, including American adolescents [23], Dutch adults [24], South-African adults [25], and with measures of mental illness such as the Brief Symptom Inventory [BSI; 18], Symptom Checklist-90-Revised [SCL-90-R; 26], and Hospital Anxiety and Depression Scale [HADS; 17].

Mental health is also referred to by the World Health Organisation as a state of well-being, specifically defining it as "a state of well-being in which the individual realises his or her own abilities, can cope with the normal stresses of life, can work productively and fruitfully, and is able to make a contribution to his or her community" [WHO; 27]. Mental health, thus, is no longer tied to the presence or absence of mental illness, but rather, pertains to the combination of (i) the individual's mental state, (ii) functioning within the social context, and (iii) the individual's psychological functioning [18, 22]. This definition and its three components are a direct reflection of our current understanding of well-being found within the literature.

Throughout the literature of well-being, two longstanding theories are highlighted, the hedonic concept of well-being and the eudaimonic concept of well-being [28–31].

The hedonic tradition was first conceptualised and views well-being as a sense of happiness, satisfaction, and interest in life [32, 33]. The three components would later be grouped under the umbrella of emotional well-being (EWB) by Keyes [34, 35].

Alternatively, the eudaimonic concept of well-being, was conceptualised later, when psychologists started to fault the narrow portrayal of well-being via the hedonic tradition, and argued that well-being ought to consider one's functioning within the social and individual

(i.e., psychological) realms [22]. Consequently, two multidimensional models of well-being; social well-being [SWB; 29], and psychological well-being [PWB; 30] were developed. The multidimensional model of SWB was developed by Keyes [29] comprises of five social dimensions. While the multidimensional model of PWB proposed by Ryff [30] comprises of six dimensions that reflect the challenges encountered as individuals strive to maximise their potential.

Incorporating both the hedonic and eudaimonic approaches, mental health can be defined as the combination of emotional, social, and psychological well-being and, therefore, be measured via these three components [19, 34]. Under the two-continua model, the absence of mental disorder is neither necessary nor sufficient to ensure a positive mental well-being, the contribution of emotional, social, and psychological well-being ought to be the primary considerations for measuring an individual's state of mental health moving forward.

## Existing measures of mental health and well-being

There are numerous well-established and validated measures of well-being, including the World Health Organisation Quality of Life Assessment [WHOQOL-100; 36], World Health Organisation Quality of Life Assessment–Brief [WHOQOL-BREF; 37], World Health Organisation-Five Well-Being Index [WHO-5; 38], Positive and Negative Affect Schedule [PANAS; 39], 12-item General Health Questionnaire [GHQ-12; 40], Satisfaction with Life Scale [SWLS; 41], and Control, Autonomy, Self-Realisation, and Pleasure Scale [CASP-19; 42].

While the above-mentioned measures have been heavily utilised, well-validated across different populations, and available in a myriad of languages, they each hold shortcomings that may deter an accurate representation and assessment of one's mental health according to the discussed definition and its three contributory factors.

Firstly, questionnaires such as the WHOQOL-100, WHOQOL-BREF, WHO-5, PANAS, SWLS, and CASP-19 measure only one or a few aspects of well-being or utilise theoretical constructs that differ from the hedonic and eudaimonic concepts of well-being. For instance, the WHOQOL-100 and WHOQOL-BREF includes physical health and environmental facets, while the WHO-5 and PANAS specifically targets the EWB component of mental health. The SWLS measures a combination of EWB and PWB aspects, and the CASP-19 utilises differing factors such as control, autonomy, self-realisation, and pleasure. Taking into consideration of the widely recognised definition of mental health by WHO [27] and the literature surrounding well-being, it is paramount for any measures of mental health to assume the three core components of mental health, emotional, social, and psychological well-being, so as to assess and present an all-encompassing representation of the individual's mental health.

Secondly, instruments such as the GHQ-12 include not only well-being, but also items of psychopathology. While mental health and mental illness have shown to be strongly related [20–22], investigations into the two-continua model have demonstrated that mental illness ought to be viewed as a distinct state, and thus, its assessment to be discrete of psychopathology [17, 18, 43].

Lastly, a clinician and researcher's choice of measure is often guided by various practical constraints, with a major consideration being administration time [44]. Instrument such as the WHOQOL-100 is rather lengthy and thus, not optimal for research or practical usage due to increased drop-out rates and fatigue [45].

Throughout the course of the current study, a new assessment tool, Well-being Numerical Rating Scales [WB-NRSs; 46], was developed and published. The instrument demonstrated numerous positives, including a concise 5 items format that comprehended most dimensions of well-being, strong psychometric properties, and the ability to detect changes in well-being

after intervention. However, while it is a promising tool, it is still new and pending of further validation studies, as currently, only one study has been conducted.

## Mental health continuum-short form

To combat the inability of existing mental health assessment tools to adequately represent the current definition of mental health, the Mental Health Continuum-Short Form [MHC-SF; 25] was developed. The MHC-SF is an abbreviated version of the 40-item Mental Health Continuum-Long Form [MHC-LF; 34].

Comprising of 14 items, each theoretical dimension of emotional (3 items), social (5 items), and psychological (6 items) well-being is operationalised using a single item (Table 1). Consequently, it ensured that the MHC-SF was theory driven and encompassed all the theoretical components of mental health.

Participants respond on a 6-point Likert scale, ranging from 0 (*Never*) to 5 (*Everyday*). Scores can be computed either continuously or categorically.

Continuous scoring would see the score of each subscale summed (i.e., EWB [range, 0 to 15], SWB [range, 0 to 25], and PWB [range, 0 to 30]), and the sum of all scores yields the overall score for well-being (range, 0 to 70). In this scenario, higher scores would indicate more positive mental health, and vice versa. No suggested cut-offs were provided [25].

The categorical scoring proposed by Keyes [34], on the other hand, captures three categorical diagnoses: languishing, moderate, and flourishing mental health. This scoring system is dependent on a specific cut-off criterion of the various well-being factors and thus, relies on the score of each subscale rather than the overall score. Languishing mental health is diagnosed when an individual reports having experienced at least one of the three EWB symptoms and at least 6 of the 11 SWB and PWB symptoms as 0 (*Never*) or 1 (*Once* or twice) within the past month. Conversely, flourishing mental health is diagnosed when an individual reports having experienced at least one of the three EWB symptoms and at least 6 of the 11 SWB and PWB

**Table 1. Dimensions of well-being and associated MHC-SF item.**

| Theoretical dimension | MHC-SF item (numbers show item order) In the past month, how often did you feel. . . |
|---|---|
| *Emotional well-being (EWB)* | |
| Happiness | 1. happy |
| Interest | 2. interested in life |
| Life satisfaction | 3. satisfied with life |
| *Social well-being (SWB)* | |
| Social contribution | 4. that you had something important to contribute to society |
| Social integration | 5. that you belonged to a community (like a social group, your neighbourhood, your city) |
| Social actualisation | 6. that our society is becoming a better place for all people |
| Social acceptance | 7. that people are basically good |
| Social coherence | 8. that the way our society works makes sense to you |
| *Psychological well-being (PWB)* | |
| Self-acceptance | 9. that you liked most parts of your personality |
| Environmental mastery | 10. good at managing the responsibilities of your daily life |
| Positive relations with others | 11. that you had warm and trusting relationships with others |
| Personal growth | 12. that you had experiences that challenged you to grow and become a better person |
| Autonomy | 13. confident to think or express your own ideas and opinions |
| Purpose in life | 14. that your life has a sense of direction or meaning to it |

symptoms as 4 (Almost every day) or 5 (Everyday) in the past month. Individuals who are neither categorised as languishing or flourishing are considered moderately mentally healthy.

The initial evaluation of the MHC-SF was conducted by Keyes, Wissing [25] with a sample of 1,050 Setswana-speaking adults in the Northwest province of South Africa. Results indicated acceptable internal consistency of the entire MHC-SF ($\alpha$ = .74), and questionable-to-acceptable range over the various subscales; EWB ($\alpha$ = .73), SWB ($\alpha$ = .59), and PWB ($\alpha$ = .67). Later evaluation studies, however, reported internal consistency of the MHC-SF total scale and its various subscales within the acceptable-to-high range ($\alpha$ = .74 to .94), in various countries such as Hong Kong, India, Japan, Malaysia, Netherlands, United Kingdom, and Vietnam [18, 47, 48].

Within the initial study, Keyes, Wissing [25] reported that the MHC-SF demonstrated convergent validity by correlating moderately to strongly with measures of subjective well-being, such as the Affectometer Positive Affect Scale ($r$ = .52, $p$ < .001) and SWLS ($r$ = .37, $p$ < .001), and measures of criteria of positive mental health, such as the Generalised Self-Efficacy Scale ($r$ = .39, $p$ < .001), and Coping Strategies Scale ($r$ = .34, $p$ < .001).

Discriminant validity of the MHC-SF has also been shown. The MHC-SF has been reported to be inversely correlated ($r$ = -.22 to -.78) with measures of mental illness, such as the BSI [18], SCL-90-R [26], and HADS [17].

With regard to the structure of the MHC-SF, Keyes, Wissing [25] conducted confirmatory factor analyses on three theoretical models: (1) one-factor model based on the assumption that all 14 items loaded on a single global factor of well-being; (2) two-factor model comprising of the two correlated dimensions of well-being: hedonic and eudaimonic well-being; and (3) three-factor model comprising of the three correlated core components of well-being: emotional, social, and psychological well-being. Results indicated that both the single and two-factor models fitted poorly, and the three-factor model was concluded to be the best-fitting model to the data, $\chi^2$ = 269.40, $df$ = 62; GFI/AGFI = .96/.94; CN = 354.50; RMSEA = .06; AIC = 345.90; $N$ = 1050 [25]. Later studies that evaluated the MHC-SF with differing populations and translated questionnaires, including reported similar results, indicating that a three-factor model fitted the best in comparison to the single and/or two-factor models, $\chi^2$ = 103.48 to 527.96, $df$ = 74; GFI = .82 to .91; AGFI = .61 to .88; RMSEA = .03 to .09; CFI = .94 to .99; SRMR = .05 to .07 [17, 18, 49, 50].

**Bifactor model of MHC-SF.** Whilst the MHC-SF was developed by Keyes [23] under the assumption of a three-factor structure (i.e., EWB, SWB, and PWB), and prior evaluation studies have demonstrated that a three-factor model to be the best fit in comparison to the single and/or two-factor models, researchers are now suggesting that the three-factor structure may be problematic [e.g., 48].

Firstly, despite the support of the three-factor model concluded in some of the previous studies [e.g., 18], the majority of studies [e.g., 17, 49] yielded fit indices that can only be considered marginally acceptable according to conventional cut-off criteria [51]. For instance, Doré, O'Loughlin [17] reported fit indices of $\chi^2$ = 527.96; $df$ = 74; $\chi^2/df$ = 7.13; RMSEA = .07; CFI = .94; SRMR = .05; TLI = .93. For this study, while the three-factor model demonstrated better fit in comparison to the one- and two-factor models, only two of the four fit indices (CFI and SRMR) were within acceptable range, and therefore, the result did not provide conclusive support to the three-factor structure of MHC-SF. Jovanović [52] highlighted the need for further investigation, so as to identify the sources of misfit, improve upon the scale, and propose, if necessary, another theoretical model that better represent the MHC-SF.

Secondly, de Bruin and du Plessis [47] have indicated that theoretically, the assumed three-factor model of MHC-SF does not align with or justify the current scoring system of the instrument. A three-factor structure of MHC-SF would theoretically translate into the calculation

and interpretation of scores separately for each of the subscale, EWB, SWB, and PWB. However, researchers, to the best of our knowledge, utilise a total MHC-SF score (i.e., continuous scoring system), which is computed by aggregating the scores across the 14 items [e.g., 18, 25]. This practice suggests the presence of a general well-being (GWB) factor, and contradicts the initial aim of the instrument, which was to measure three relatively distinct components of well-being [52]. Several evaluation studies, including Jovanović's [52] and Petrillo et al.'s [58] studies, have included a second-order factor in their analyses, revealing the presence of a GWB factor amongst the 14 items.

A second-order factor analysis, however, does not allow for direct comparisons of strengths between the GWB factor and the group factors (i.e., EWB, SWB, and PWB), and thus is not able to measure any contribution of unique variance by the various subscales [47, 52]. Consequently, numerous researchers including de Bruin and du Plessis [47], Jovanović [52], and Żemojtel-Piotrowska, Piotrowski [48] have incorporated a bifactor model in their analyses, where each item loaded on a general factor (i.e., GWB) and simultaneously on one of the uncorrelated group factors (i.e., EWB, SWB, and PWB). Bifactor model analysis has been widely utilised in understanding the structures of multidimensional constructs measures such as self-esteem [53], depression [54], and intelligence [55]. Importantly, it allows for the separation of the GWB dimension from the group factors (i.e., EWB, SWB, and PWB), enabling investigations into the proportion of variance contributed by each dimension, dimensionality of the data, and reliability of the GWB dimension and subscales [47, 48, 56].

In a cross-cultural evaluation study of MHC-SF's structure involving 38 countries ($N$ = 8,066), Żemojtel-Piotrowska, Piotrowski [48] reported that in all 38 nations, the bifactor model fitted better than the three-factor model, with 16 countries demonstrating good fit, and acceptable fit in all others, except Kenya and Iran, where the fit was marginal. Excluding the two countries which demonstrated marginal fit, the fit indices were as follows: $\chi^2$ = 74.20 to 155.31, $df$ = 63; RMSEA = .32 to .77; CFI = .90 to .98; SRMR = .37 to .63. In addition, the indices evaluating the reliability and dimensionality of the MHC-SF also demonstrated prominent results in all samples. The Macdonald's ω reliability coefficients ranged from .82 to .95 for the GWB factor and .57 to .92 for the subscales. In addition, the Macdonald's $\omega_H$ coefficient ranged from .56 to .87 ($M$ = .80), and the explained common variance (ECV) index of the GWB factor ranged from .40 to .76 ($M$ = .66), while the mean Macdonald's $\omega_S$ coefficients for the various subscales are as follows: .29 for EWB; .31 for SWB; and .12 for PWB.

Studies, including an Australian sample [57], who have incorporated a bifactor model as part of their structural analysis of the MHC-SF have also found similar results; good and better fit over the other proposed models, good reliabilities coefficients, and unidimensionality of the data [e.g., 47, 48, 52].

Overall, the results indicate that a bifactor model would structurally better represent the MHC-SF, and the combination of a high ECV and high $\omega_H$ coefficient indicates unidimensionality of the model and supports the computation and interpretation of a total MHC-SF score via the continuous scoring system.

**Aims and hypotheses.** The MHC-SF is seemingly an instrument that has the ability to address the limitations of the other measures of mental health and has been translated, validated, and utilised across many different languages and countries, such as Chinese [49], Italian [58], Serbian [52], and Polish [59]. In addition, given the importance of mental health, and its relationship to mental disorders, it is paramount that such a promising instrument is properly evaluated and validated in as many different contexts as possible. To the best of our knowledge, there has only been one study conducted with an Australia sample [57], and none with a Singapore sample. Therefore, the primary aim of the current study was to examine the psychometric properties of the MHC-SF within the Singapore and Australia context. The second aim was to

provide further clarification on the structural model of the MHC-SF and better inform the computation and interpretation of its scores. Thus, with consideration of the discussed literature and aims of the current study, the hypotheses of the current study (applicable to both samples) were as follows:

H1, A bifactor model of MHC-SF would demonstrate good, and comparatively better fit than the other models (i.e., one-factor, two-factor, and three-factor models).

H2, Results of the bifactor models would demonstrate a combination of high ECV ($\geq$ .70) and high Macdonald's $\omega_H$ coefficient ($\geq$ .80), thus providing support for unidimensionality and interpretation for a total MHC-SF score.

H3, MHC-SF total scale and various subscales would demonstrate high internal consistency reliability (Cronbach's $\alpha$ and Macdonald's $\omega$ reliability coefficients $\geq$ .80).

H4, MHC-SF's total and subscales scores would demonstrate moderate to strong positive correlations with scores in WHO-5, which measures an individual's current mental well-being based on positive mood, vitality, and general interest (Pearson's $r \geq$ .30), demonstrating concurrent validity.

H5, MHC-SF's total and subscales scores would demonstrate moderate to strong positive correlations with the participants' self-rated level of mental health (Pearson's $r \geq$ .30), demonstrating criterion validity.

H6, MHC-SF's total and subscales scores would demonstrate moderate to strong negative correlations with both the depression and anxiety subscales of HADS (Pearson's $r \geq$ -.30), demonstrating discriminant validity.

## Methods

### Design and participants

The current study employed a cross-sectional and correlational research design. An individual was eligible for the study if he/she was above the 18 years old and a citizen or permanent resident (PR) of Singapore or Australia. Participants were either recruited via convenience sampling through the James Cook University's (Singapore and Australia campuses) research participation program, to which upon completion, received 1 research credit point, or have encountered the study through snowball techniques, primarily distributed via social networks.

A total of 588 participants were recruited, but 31 records were omitted due to missing data. Of which, 410 participants were recruited via the university's research participation program and 147 participants via the snowball technique. The data was then separated into two groups: the Singapore sample, consisting of 299 Singaporeans or PRs; and the Australia sample, consisting of 258 Australians or PRs. The characteristics of participants are shown in Table 2.

### Measures

**Mental health continuum-short form.** The MHC-SF [25] is a 14-item self-reported measure of mental health, investigating the respondent's emotional (3 items), social (5 items), and psychological (6 items) well-being. Participants responded on a 6-point Likert scale, ranging from 0 (*Never*) to 5 (*Everyday*). Scores can be computed either continuously or categorically. The current study predominantly utilised the continuous scoring system; the scores of each subscale summed (i.e., EWB [range, 0 to 15], SWB [range, 0 to 25], and PWB [range, 0 to 30]),

**Table 2. Characteristics of participants.**

| Characteristics | Singapore sample (N = 299) | | Australia sample (N = 258) | |
|---|---|---|---|---|
| Age | | | | |
| Mean (SD) | 24.26 | (6.13) | 23.95 | (8.66) |
| Range | 18 to 63 | | 18 to 65 | |
| Gender, n (%) | | | | |
| Male | 120 | (40.10%) | 76 | (30.00%) |
| Female | 177 | (59.20%) | 178 | (69.00%) |
| Other/Do not wish to disclose | 2 | (.70%) | 2 | (1.00%) |
| Ethnicity, n (%) | | | | |
| Chinese | 247 | (82.60%) | 3 | (1.20%) |
| Malay | 10 | (3.30%) | 1 | (.40%) |
| Indian | 28 | (9.40%) | 1 | (.40%) |
| Caucasian | - | | 202 | (78.20%) |
| Other | 14 | (4.70%) | 51 | (19.80%) |
| Citizenship, n (%) | | | | |
| Citizens | 279 | (93.30%) | 242 | (93.80%) |
| PRs | 20 | (6.70%) | 16 | (6.20%) |
| Marital status, n (%) | | | | |
| Single (never married) | 255 | (85.20%) | 155 | (60.00%) |
| Married | 28 | (9.40%) | 13 | (5.00%) |
| Partnered | 16 | (5.40%) | 67 | (26.00%) |
| Separated/divorced | - | | 17 | (6.60%) |
| Widowed | - | | 3 | (1.20%) |
| Other | - | | 3 | (1.20%) |
| Education, n (%) | | | | |
| Primary or less | - | | 4 | (1.60%) |
| Secondary | 3 | (1.00%) | 132 | (51.20%) |
| Pre-University or vocational | 154 | (51.50%) | 64 | (24.80%) |
| University or above | 141 | (47.20%) | 57 | (22.00%) |
| Special education | 1 | (.30%) | 1 | (.40%) |
| Employment status, n (%) | | | | |
| Student | 190 | (63.50%) | 199 | (46.10%) |
| Employed | 89 | (29.80%) | 116 | (45.00%) |
| Unemployed | 10 | (3.40%) | 13 | (5.00%) |
| Self-employed | 6 | (2.00%) | 4 | (1.60%) |
| Retired | 1 | (.30%) | 1 | (.40%) |
| Other | 3 | (1.00%) | 5 | (1.90%) |
| Population groups, n (%) | | | | |
| Adults with no disabilities or mental health issues | 250 | (83.60%) | 172 | (66.70%) |
| Adult with disabilities | 8 | (3.00%) | 14 | (5.40%) |
| Adults recovering from mental health issues | 40 | (13.40%) | 72 | (27.90%) |

and totality of the 14 items for the overall well-being score (range, 0 to 70). Higher scores would indicate more positive mental health, and vice versa.

The initial evaluation of the MHC-SF by Keyes, Wissing [25] reported acceptable internal consistency of the entire MHC-SF ($\alpha$ = .74), and questionable-to-acceptable range over the various subscales; EWB ($\alpha$ = .73), SWB ($\alpha$ = .59), and PWB ($\alpha$ = .67). While later studies, such as de Bruin and du Plessis [47], Lamers, Westerhof [18], and Żemojtel-Piotrowska, Piotrowski

[48] reported internal consistency of the MHC-SF total scale and its various subscales within the acceptable-to-high range (α = .74 to .94). The MHC-SF has good convergent validity [25] and good discriminant validity [17, 18, 26].

**World health organisation-five well-being index.** The WHO-5 [38] is a 5-item self-reported measure of current mental well-being based on positive mood, vitality, and general interest. Respondents rate their degree of agreement with statements that describe how they felt in the last two weeks (e.g., "I have felt calmed and relaxed"), on a 6-point Likert scale, ranging from 0 (*At no time*) to 5 (*All of the time*). Aggregation across the five items would result in a total score ranging from 0 to 25, with 0 representing worst possible and 25 representing best possible quality of life.

Studies have demonstrated the WHO-5 to have high internal consistency, with Cronbach's α coefficients above .80 [60, 61]. Cronbach's α in the current study was .93 and .90 for the Singapore and Australia sample respectively, indicative of excellent internal consistency. In addition, studies have reported the WHO-5 to have good construct validity [62, 63], good predictive validity [64], and good convergent validity [65, 66].

**Hospital anxiety and depression scale.** The HADS [67] is a 14-item brief self-reported questionnaire that measures symptoms of anxiety and depression, with seven items measuring each subscale, HADS-Anxiety and HADS-Depression. Respondents rate their degree of agreement with statements that describe how they felt in the last seven days (e.g., "I feel as if I am slowed down"), on a 4-point Likert scale. Aggregating the scores for each subscale would result in a subscale score ranging from 0 to 21, with 0 to 7 representing normal range, 8 to 10 representing borderline abnormal range, and 11 to 21 representing abnormal range.

HADS has demonstrated acceptable-to-high internal consistency across the literature; with Cronbach's α ranging from .68 to .93 for the anxiety, and .67 to .90 for depression subscales [68]. Both subscales demonstrated good internal consistency for the current study; Cronbach's α = .89 and .85 for the anxiety subscale for the Singapore and Australia samples, respectively; and Cronbach's α = .87 and .77 for the depression subscale for the Singapore and Australia samples, respectively. Studies have also identified the HADS to have good factorial validity [69] and discriminant validity [68].

**Self-rated mental health question.** A single self-rated question inquiring about the respondent's evaluation of their own mental health was included ("Please rate your averaged level of mental health over the past month"). Participants responded on a 11-point Likert scale, ranging from 0 (*Poor*) to 10 (*Excellent*). This question was included as a mean of comparison to investigate the criterion validity of the MHC-SF.

## Procedure

All procedures in the current study were approved and in accordance with the ethical standards of James Cook University's Human Research Ethics Committee (Ethics number: #H7889).

The questionnaires were administered online via Qualtrics. The participants were free to complete the survey on their own time, it was expected that the survey took no longer than 15 minutes to complete. Participants were informed they were free to withdraw while responding, but they could not withdraw their data once submitted because the data were collected anonymously and could not be identified.

When the participants first clicked on the link to the study, they were presented with an Information Sheet briefly explaining the nature of the study and the questionnaires they will be required to complete. Informed consent was acknowledged when the participants continued past the Information Sheet. Subsequently, they were invited to complete the Background

Information Form, the MHC-SF, the WHO-5, and the HADS. Finally, they were presented with a Feedback Form reiterating the nature of the study and thanking them for their participation.

Lastly, participants who were students of JCU and signed up via the research participation system would be granted one credit point towards their required research participation.

### Statistical analysis

Analyses were conducted for each sample separately. Pearson's correlations coefficients were conducted using the SPSS version 21, confirmatory factor analyses (CFA) were conducted using the AMOS version 21, and the ECV and MacDonald's ω indices were calculated using the Bifactor Indices Calculator by Dueber. All statistical analyses were conducted with alpha level set at .05.

Confirmatory factor analyses (CFA) were employed to find the best fitting model and test structural validity of the MHC-SF. The models were identified by fixing the latent factor variances to 1 and freely estimating the factor loadings [70]. Knowing the limitations of the chi-square test of overall model fit [71], the following fit indices and their respective recommended cut-offs by Hu and Bentler [51] were utilised: $\chi^2/df$ with a value of less than 3 being adequate; root mean square error of approximation (RMSEA) with a value of .06 or less being indicative of a good fit; standardised root mean square residual (SRMR) with a value of .08 or below as indications of good fit; and comparative fit index (CFI) with a value greater than .90 being acceptable fit and .95 and above being good fit.

MacDonald's $\omega_H$ and $\omega_S$ coefficients were caluclated to identify any unique variance of well-being contributed by the GWB dimension and the various subscales [56]. The MacDonald's $\omega_H$ coefficient will report the share of total variance contributed by the GWB factor, while the MacDonald's $\omega_S$ coefficient will identify the unique share of variance contributed by each subscale, excluding the contribution of the GWB factor. According to Reise, Bonifay [72], when the MacDonald's $\omega_H$ coefficient is > .80, the model be considered unidimensional and consequently, justifying the current aggregating scoring system of the MHC-SF employed by all researchers. ECV coefficient was also calculated to measure the relative strength of the GWB factor to the group factors. According to Rodriguez, Reise [73], an ECV value of .70 and above would indicate a strong general factor and that the common variance is essentially unidimensional. Therefore, a high ECV value, in tandem with a high MacDonald's $\omega_H$ coefficient, would provide strong support to the unidimensionality of the model and the use of the continous scoring system.

Due to Cronbach's α limitations with multidimensional measures [74], the current study included the use of MacDonald's ω coefficient to test for the internal consistency reliability of MHC-SF. The MacDonald's ω coefficient reflects the proportion of total item variance explained by the combined contribution of the GWB factor and the three group factors. As such, the MacDonald's ω coefficient, akin to Cronbach's α, allows the reliability of the GWB dimension and the three subscales to be evaluated. According to Reise, Bonifay [72], the Mac-Donald's ω coefficient can be interpreted similiarly to other popular indices of reliably, therefore, the current study employed the recommendations by Cortina [75]: .70 as acceptable, and .80 or higher as good. Similarly, Cronbach's α was calculated in tandem and the same recommended cut-offs were employed.

Concurrent validity of the MHC-SF was investigated via its correlations with other validated measures of mental health. For the current study, a Pearson's correlation between the MHC-SF and WHO-5, which measures the individual's current mental well-being based on

positive mood, vitality, and general interest was conducted, and a correlation coefficient of .30 or above would indicate good convergent validity [76].

Criterion validity of the MHC-SF was tested via a Pearson's bivariate correlation of the MHC-SF with the self-rated mental health question ("Please rate your averaged level of mental health over the past month"). Similarly, a correlation coefficient of .30 or above would be indicative of good criterion validity [76].

Indication of discriminant validity was based on the current understanding and results of prior studies of the two-continua model of mental health and mental illness. According to Doré, O'Loughlin [17], a negative moderate to high correlation coefficient of the MHC-SF and measure of psychopathology (e.g., HADS) would be supportive of the two-continua model and indicative of discriminant validity. Thus, the current study conducted a Pearson's bivariate correlation between MHC-SF and the two subscales of HADS which measures symptoms of anxiety and depression, and employed the cut-off of |.30| or above as indication of moderate correlation, and |.50| or greater as strong correlation [76].

## Results

The descriptive statistics, continuous scores and categorical results of the MHC-SF for both the Singapore and Australia samples are presented in Table 3.

### Structural validity

Based on the theoretical underpinnings and results of prior studies (e.g., de Bruin & du Plessis, 2015; Hides et al., 2016; Jovanović, 2015; Żemojtel-Piotrowska et al., 2018), the current study tested four different CFA models of the MHC-SF: (a) a single-factor model, where all 14 items loaded on one general dimension of well-being (Fig 1); (b) a two-factor model with two correlated dimensions of well-being (i.e., hedonic well-being, comprising of items 1 to 3, and eudaimonic well-being, comprising of items 4 to 14; Fig 2); (c) a three-factor model with three correlated dimensions of well-being (i.e., EWB consisting of items 1 to 3, SWB consisting of items 4 to 8, and PWB consisting of items 9 to 14; Fig 3); and (d) a bifactor model with a GWB factor and three uncorrelated group factors capturing the specific variance of EWB, SWB, and PWB (Fig 4). A single second-order factor model was not tested because a hierarchical model with three first-order factors is statistically equivalent to the correlated three-factor model.

The fit indices across both samples for all four models are shown in Table 4. Across both samples, the single-factor model did not fit the data well. With the Singapore sample, only the SRMR fit index (.80) can be considered a good fit, while the Australia sample had all four fit indices lying outside of acceptable ranges.

**Table 3. Descriptive statistics, continuous scores, and categorical results of the MHC-SF.**

| Country | Well-being | M (SD) | Mean sum of score (SD) / Maximum possible score | Languishing (%) | Moderate (%) | Flourishing (%) |
|---|---|---|---|---|---|---|
| Singapore (N = 299) | GWB | 2.91 (1.30) | 40.74 (14.00) / 70 | 10.00% | 54.50% | 35.50% |
| | EWB | 2.60 (1.39) | 9.31 (2.99) / 15 | | | |
| | SWB | 3.10 (1.11) | 12.98 (5.75) / 25 | | | |
| | PWB | 3.07 (1.27) | 18.44 (6.30) / 30 | | | |
| Australia (N = 258) | GWB | 2.99 (1.40) | 41.88 (13.90) / 70 | 5.80% | 60.90% | 33.30% |
| | EWB | 3.46 (1.16) | 10.38 (3.10) / 15 | | | |
| | SWB | 2.50 (1.46) | 12.48 (5.90) / 25 | | | |
| | PWB | 3.17 (1.34) | 19.02 (6.27) / 30 | | | |

*Note.* M = mean, SD = standard deviation, GWB = general well-being, EWB = emotional well-being, SWB = social well-being, PWB = psychological well-being.

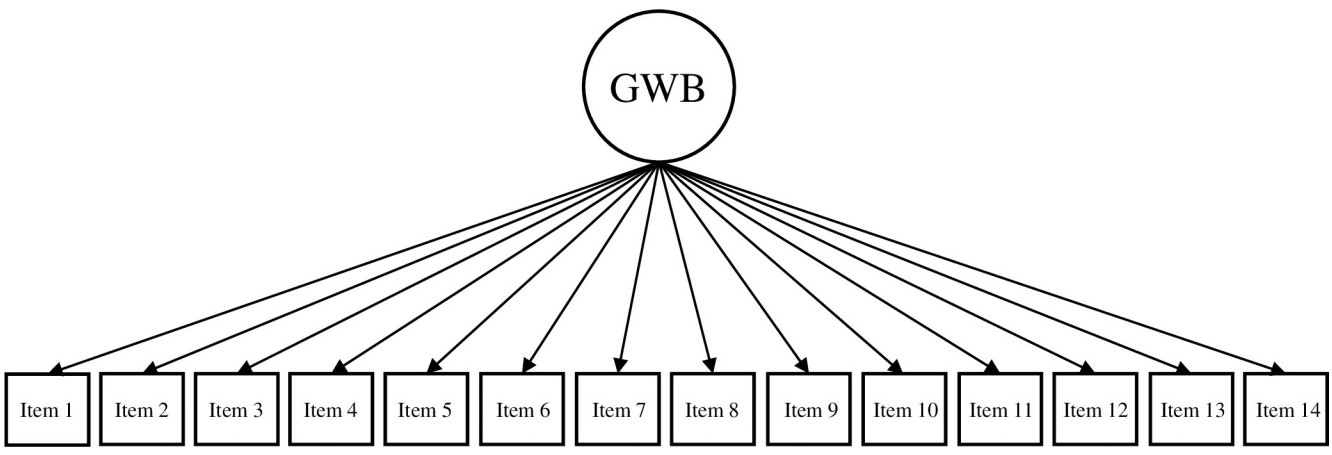

**Fig 1. Single-factor model of the MHC-SF.** *Note.* GWB = general well-being.

The two-factor model, albeit demonstrated improvements in fitting compared to the single-factor model across both samples, could not be considered acceptable. The Singapore sample reported two fit indices outside of their respective acceptable ranges, with only the SRMR (.07) and CFI (.92) showing indications of a good and acceptable fit respectively. The Australia sample fitter poorer comparatively, with three of the four fit indices lying outside of acceptable ranges, with only the CFI fit index of .91 being marginally acceptable.

Based on the combination of indices, the three-factor model for both samples, whilst demonstrating comparatively better fit than the prior two models, could not be confidently concluded to be within acceptable ranges. Both samples had only two of the four fit indices within the recommended ranges. Singapore sample reported SRMR of .07 and CFI of .94, indicative of good and acceptable fit respectively, while the Australia sample demonstrated adequate $\chi^2$/*df* value of 2.65 and good CFI value of .95.

Unlike the previous three models, the bifactor model for both Singapore and Australia samples can be conclusively concluded to have demonstrated good and excellent fit to the data respectively. The bifactor model for the Singapore sample reported the best fit to its data, with

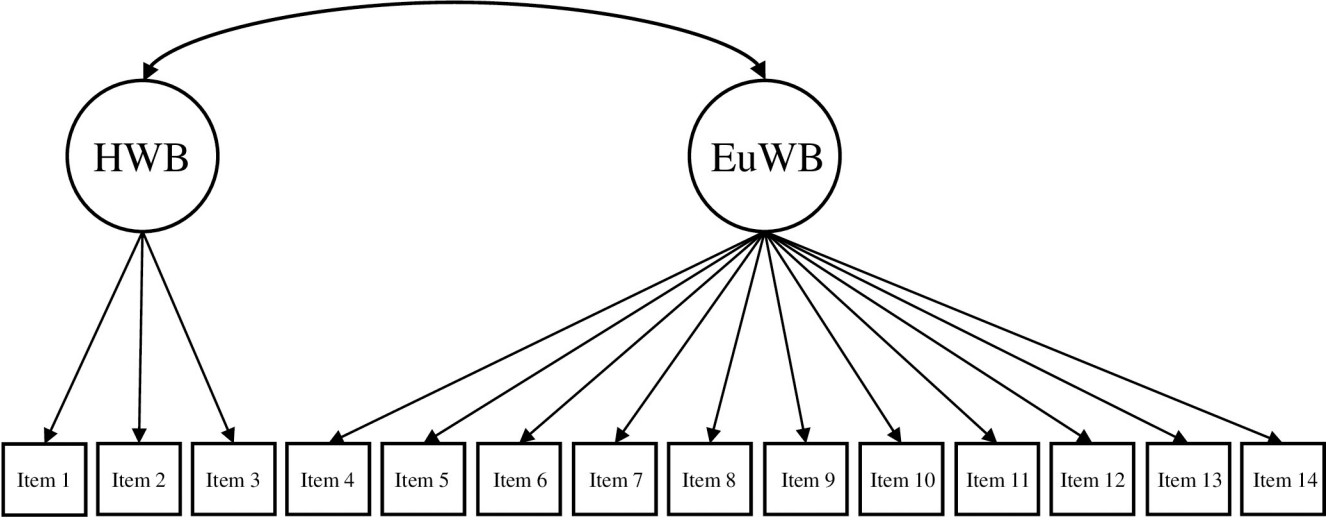

**Fig 2. Two-factor model of the MHC-SF.** *Note.* HWB = hedonic well-being, EuWB = eudaimonic well-being.

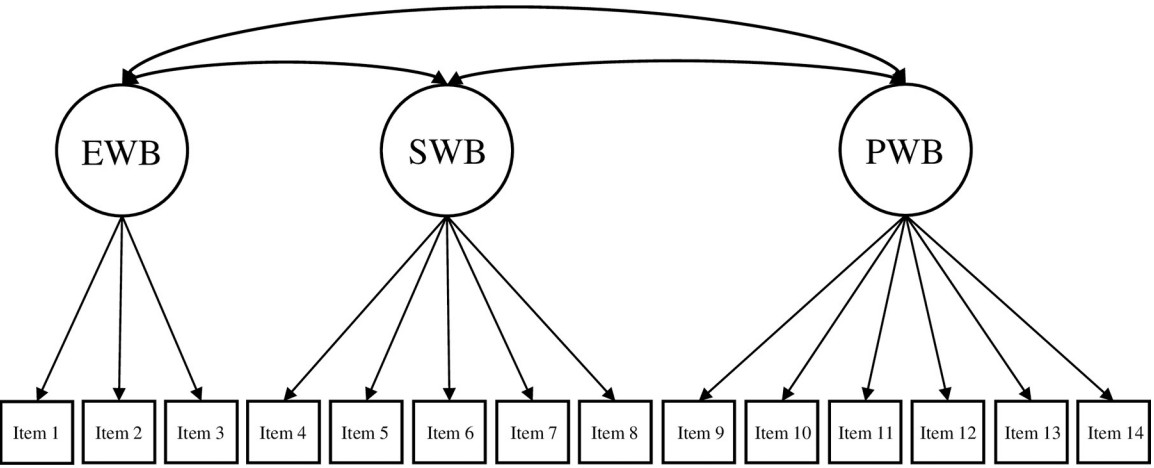

**Fig 3. Three-factor model of the MHC-SF.** *Note.* EWB = emotional well-being, SWB = social well-being, PWB = psychological well-being.

only the RMSEA fit index (.08) marginally outside of its recommended range. Similarly, the bifactor model for the Australia sample reported the best fit to its data, with all fit indices well within their recommended ranges.

Following the fit indices, the standardised factor loadings, MacDonald's $\omega_H$ and $\omega_S$ coefficients, and ECV of the bifactor model for Singapore sample (Table 5) and Australia sample (Table 6) were investigated.

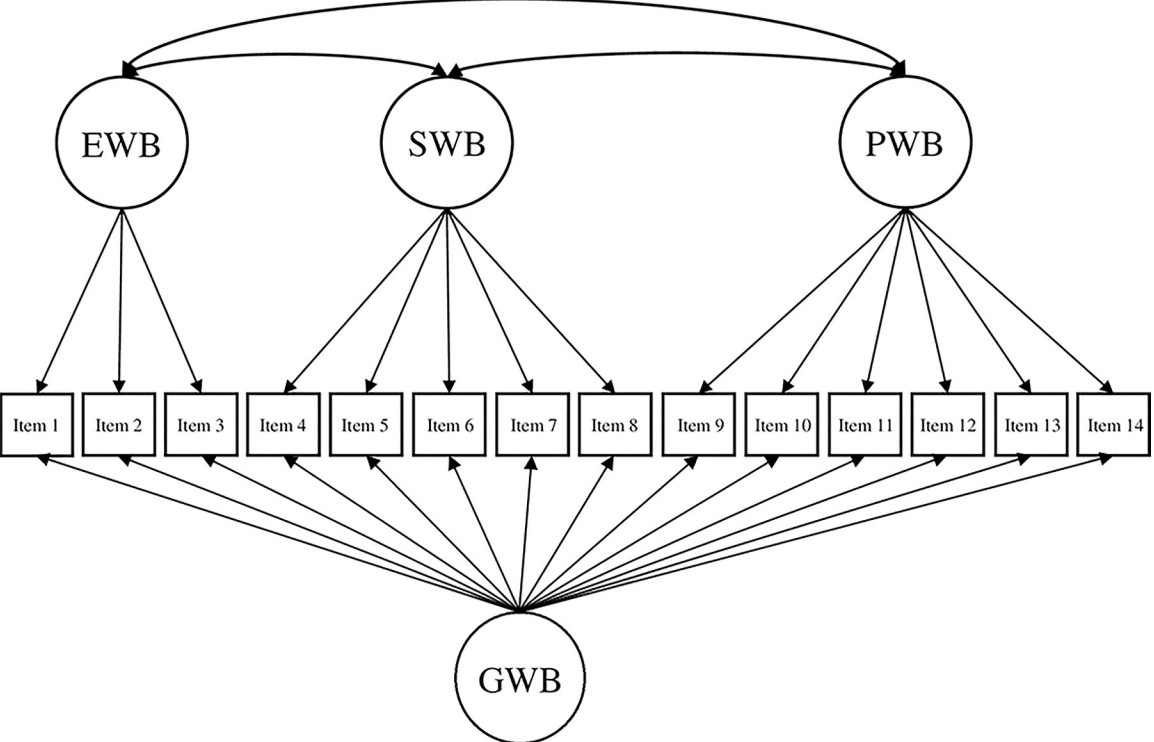

**Fig 4. BiFactor model of the MHC-SF.** *Note.* GWB = general well-being, EWB = emotional well-being, SWB = social well-being, PWB = psychological well-being.

**Table 4. Confirmatory factor analysis fit statistics for the MHC-SF.**

| Country | Model | $\chi^2$ (*df*) | $\chi^2$/*df* | RMSEA (90% CI) | SRMR | CFI |
|---|---|---|---|---|---|---|
| Singapore (*N* = 299) | Single factor | 464.03 (77) * | 6.03 | .13 (.12 - .14) | .08 | .89 |
| | Two correlated factors | 339.61 (76) * | 4.47 | .11 (.01 - .12) | .07 | .92 |
| | Three correlated factors | 263.04 (74) * | 3.56 | .09 (.08 - .11) | .07 | .94 |
| | Bifactor | 177.91 (63) * | 2.82 | .08 (.07 - .09) | .05 | .97 |
| Australia (*N* = 258) | Single factor | 404.24 (77) * | 5.25 | .13 (.12 - .14) | .11 | .86 |
| | Two correlated factors | 284.37 (76) * | 3.74 | .10 (.09 - .12) | .10 | .91 |
| | Three correlated factors | 195.93 (74) * | 2.65 | .08 (.07 - .09) | .10 | .95 |
| | Bifactor | 98.75 (63) ** | 1.57 | .05 (.03 - .06) | .05 | .99 |

*Note.*

* $p$ = .001

** $p$ = .003

$\chi^2$ = chi square, *df* = degrees of freedom, RMSEA = root mean square error of approximation, CI = confidence intervals, SRMR = standardised root mean square residual, CFI = comparative fit index.

The bifactor model for the Singapore sample produced a strong GWB factor, with standardised factor loadings ranging from .69 to .88. Notably, all items had higher loadings on the GWB factor than on their respective specific factors. In addition, whilst all EWB items had significant loadings on both the GWB and EWB factors, two SWB items (items 4 [social contribution] and 5 [social integration]) and most of PWB items (excluding item 9 [self-acceptance]) had low loadings on its specific factors. The results provided a brief indication that the

**Table 5. Standardised factor loadings, Macdonald's omega hierarchical and specific coefficients, and explained common variance of the bifactor model of the MHC-SF (Singapore sample).**

| Singapore Sample (*N* = 299) | | | | |
|---|---|---|---|---|
| Item | GWB | EWB | SWB | PWB |
| 1. happy | .72 | .46 | | |
| 2. interested in life | .78 | .31 | | |
| 3. satisfied with life | .81 | .41 | | |
| 4. that you had something important to contribute to society | .69 | | .25 | |
| 5. that you belonged to a community (like a social group, your neighbourhood, your city) | .81 | | .16 | |
| 6. that our society is becoming a better place for all people | .71 | | .58 | |
| 7. that people are basically good | .72 | | .37 | |
| 8. that the way our society works makes sense to you | .68 | | .38 | |
| 9. that you liked most parts of your personality | .84 | | | .42 |
| 10. good at managing the responsibilities of your daily life | .75 | | | .15 |
| 11. that you had warm and trusting relationships with others | .80 | | | .14 |
| 12. that you had experiences that challenged you to grow and become a better person | .69 | | | .13 |
| 13. confident to think or express your own ideas and opinions | .78 | | | .20 |
| 14. that your life has a sense of direction or meaning to it | .88 | | | -.12 |
| Omega hierarchical coefficient (MacDonald's $\omega_H$) | .92 | | | |
| Proportion of explained common variance (ECV) | .85 | | | |
| Omega specific coefficient (MacDonald's $\omega_S$) | | .19 | .17 | .03 |

*Note.* The order of the items corresponds with the ordering of the item labels in Keyes, Wissing [25]. GWB = general well-being, EWB = emotional well-being, SWB = social well-being, PWB = psychological well-being.

**Table 6. Standardised factor loadings, Macdonald's omega hierarchical and specific coefficients, and explained common variance of the bifactor model of the MHC-SF (Australia sample).**

| Australia Sample (N = 258) | | | | |
|---|---|---|---|---|
| Item | GWB | EWB | SWB | PWB |
| 1. happy | .66 | .45 | | |
| 2. interested in life | .80 | .41 | | |
| 3. satisfied with life | .78 | .46 | | |
| 4. that you had something important to contribute to society | .72 | | .13 | |
| 5. that you belonged to a community (like a social group, your neighbourhood, your city) | .73 | | .23 | |
| 6. that our society is becoming a better place for all people | .64 | | .69 | |
| 7. that people are basically good | .60 | | .50 | |
| 8. that the way our society works makes sense to you | .60 | | .47 | |
| 9. that you liked most parts of your personality | .79 | | | -.07 |
| 10. good at managing the responsibilities of your daily life | .68 | | | .07 |
| 11. that you had warm and trusting relationships with others | .74 | | | .14 |
| 12. that you had experiences that challenged you to grow and become a better person | .64 | | | .77 |
| 13. confident to think or express your own ideas and opinions | .70 | | | .11 |
| 14. that your life has a sense of direction or meaning to it | .82 | | | .07 |
| Omega hierarchical coefficient (MacDonald's $\omega_H$) | .89 | | | |
| Proportion of explained common variance (ECV) | .76 | | | |
| Omega specific coefficient (MacDonald's $\omega_S$) | | .23 | .24 | .05 |

*Note.* The order of the items corresponds with the ordering of the item labels in Keyes, Wissing [25]. GWB = general well-being, EWB = emotional well-being, SWB = social well-being, PWB = psychological well-being.

variances of all EWB items were likely spilt between GWB factor and its specific factor, while the variances of some SWB and PWB items were likely explained by the GWB factor.

In addition, the model highlighted the dominant strength of the GWB factor. The MacDonald's $\omega_H$ coefficient, which reflects the proportion of total variance explained by the general factor was .92, indicating the significant contribution of the GWB factor to the model. Furthermore, the ECV index of the GWB factor was .85, which further supported the substantial contribution of the GWB factor in the true score variance. On the other hand, the MacDonald's $\omega_S$ coefficients of the various group factors (EWB = .19, SWB = .17, PWB = .03), demonstrated their minimal contribution to the true score variance when excluding the contribution of the general factor. Notably, the PWB factor reported a comparatively smaller amount of unique variance (.03), which suggested that the variance it captures is better represented by the combination of all three subscales of MHC-SF, whereas the EWB (.19) and SWB (.17) factors demonstrated more distinct standalone contributions.

In combination, results of the MacDonald's $\omega_H$ and $\omega_S$ coefficients as well as the ECV index strongly suggested that majority of the variance captured by the MHC-SF were shared by the three scales. Accordingly, the model can be identified as unidimensional, therefore suggesting that the GWB factor of MHC-SF to be most appropriate as an indicator of overall mental health in Singapore.

Similar to the model obtained in Singapore, the bifactor model of the Australia sample produced a strong GWB factor, with standardised factor loadings ranging from .60 to .80. All items except item 6 (SWB; [social actualisation]) and item 12 (PWB; [personal growth]) reported higher loadings on the GWB factor than on their respective specific factors. In addition, whilst all EWB items had significant loadings on both the GWB and EWB factors, two SWB items (items 4 [social contribution] and 5 [social integration]) and most of PWB items

(excluding item 12 [personal growth]) had low loadings on its specific factors. This result, provided a brief indication that the variances of all EWB items were likely spilt between GWB factor and its specific factor, while the variances of some SWB and PWB items were likely explained by the GWB factor.

Results of the MacDonald's $\omega_H$ and $\omega_S$ coefficients and ECV index for the Australia model were highly similar to that of the Singapore's sample. The model highlighted the dominant strength of the GWB factor, with high MacDonald's $\omega_H$ coefficient (.89) and high ECV index (.76). Likewise, the MacDonald's $\omega_S$ coefficients of the various group factors were low, with the PWB factor reporting comparatively lower amount of unique variance (.05), while the EWB (.23) and SWB (.24) factors demonstrated more distinct standalone contributions.

Akin to the Singapore model, results of the Australia sample suggested that majority of the variance captured by the MHC-SF were shared by the three scales. Accordingly, the model can be identified as unidimensional, suggesting that the GWB factor of MHC-SF to be most appropriate as an indicator of overall mental health in Australia.

## Internal consistency reliability

Table 7 reports the Cronbach's α and MacDonald's ω reliability coefficients of MHC-SF for both samples.

Cronbach's α coefficients was .95 (Singapore) and .94 (Australia) for the total MHC-SF score, and ranged from .87 to .92 for the three subscales. The MacDonald's ω reliability coefficients, which demonstrates the proportion of true score variance while accounting for all factors, was .97 (Singapore) and .96 (Australia) for the GWB index, and ranged from .90 to .92 for the three subscales. Across both countries and assessments of reliability, the MHC-SF demonstrated high internal consistency across the total scale and with its various subscales.

## Concurrent validity

The MHC-SF measuring an individual's well-being across three dimensions (emotional, social, and psychological) was compared against the WHO-5 which measures an individual's current mental well-being based on positive mood, vitality, and general interest. Pearson's correlations of the MHC-SF total scale and subscales with the WHO-5 were examined (Table 8). The MHC-SF total scale and subscales across both countries demonstrated strong positive correlations with the WHO-5, ranging from $r = .66$ to $.86$, and are indicative of concurrent validity for the MHC-SF.

**Table 7. Cronbach's α and MacDonald's ω reliability coefficients of MHC-SF.**

| Reliability coefficient | Singapore sample ($N = 299$) | Australia sample ($N = 258$) |
|---|---|---|
| Cronbach's α | | |
| Total MHC-SF score | .95 | .94 |
| Emotional well-being score | .89 | .89 |
| Social well-being score | .90 | .87 |
| Psychological well-being score | .92 | .88 |
| MacDonald's ω reliability | | |
| General well-being factor | .97 | .96 |
| Emotional well-being factor | .90 | .90 |
| Social well-being factor | .91 | .90 |
| Psychological well-being factor | .92 | .90 |

**Table 8. Pearson's correlations of MHC-SF and WHO-5.**

| | Singapore sample (*N* = 299) | Australia sample (*N* = 258) |
|---|---|---|
| **MHC-SF Scale and Subscales** | **WHO-5** | |
| Total MHC-SF score | .86 | .79 |
| Emotional well-being scale | .83 | .79 |
| Social well-being scale | .73 | .66 |
| Psychological well-being scale | .84 | .74 |

*Note*. All correlations are significant at the .01 level (two-tailed)

## Criterion validity

Pearson's correlations of the MHC-SF total scale and subscales with the self-rated mental health question ("Please rate your averaged level of mental health over the past month") were examined (Table 9). The MHC-SF total scale and subscales across both countries demonstrated moderate to strong positive correlations with the self-rated question, ranging from *r* = .52 to .72, and are indicative of criterion validity for the MHC-SF.

## Discriminant validity

Table 10 shows the Pearson's correlations between MHC-SF total scale and subscales and both of HADS subscales for both samples. HADS anxiety and depression subscales negatively correlated with MHC-SF's total scale and its three subscales, ranging from -.45 to -.71. Higher presence of psychopathology symptoms was moderately linked to lower levels of mental well-being across all dimensions. This result provided support to the two-continua model of mental health and mental illness, where both constructs are thought to be highly related yet distinct. In addition, the result is indicative of discriminant validity for the MHC-SF.

## Discussion

The current study aimed to evaluate the psychometric properties of the MHC-SF in Singapore and Australia, to clarify on the structural model of the MHC-SF, as well as to better inform the computation and interpretation of its scores.

Similar to prior studies that have included a bifactor model within their consideration [e.g., 47, 48, 52], results of the current study demonstrated that a bifactor model of the MHC-SF yielded the best fit to the data. The bifactor model with one GWB factor and three specific factors of EWB, SWB, and PWB, reported good and excellent fittings to the Singapore and Australia samples. Unlike the bifactor model, the fit indices of the single, two, and three-factor models of the MHC-SF could not be confidently concluded to be within acceptable ranges.

**Table 9. Pearson's correlations of MHC-SF and self-rated question on mental health.**

| | Singapore sample (*N* = 299) | Australia sample (*N* = 258) |
|---|---|---|
| **MHC-SF Scale and Subscales** | **"Please rate your averaged level of mental health over the past month"** | |
| Total MHC-SF score | .72 | .62 |
| Emotional well-being scale | .70 | .65 |
| Social well-being scale | .62 | .52 |
| Psychological well-being scale | .71 | .57 |

*Note*. All correlations are significant at the .01 level (two-tailed)

**Table 10. Pearson's correlations of MHC-SF and HADS.**

| MHC-SF Scale and Subscales | Singapore sample (*N* = 299) | | Australia sample (*N* = 258) | |
|---|---|---|---|---|
| | HADS-A | HADS-D | HADS-A | HADS-D |
| Total MHC-SF score | -.67 | -.71 | -.52 | -.71 |
| Emotional well-being scale | -.58 | -.68 | -.45 | -.70 |
| Social well-being scale | -.62 | -.61 | -.45 | -.58 |
| Psychological well-being scale | -.65 | -.69 | -.50 | -.69 |

*Note*. All correlations are significant at the .01 level (two-tailed). HADS-A = Hospital Anxiety and Depression Scale-Anxiety Subscale, HADS-D = Hospital Anxiety and Depression Scale-Depression Subscale.

The results of the current study do not support the three factor structure of the MHC-SF proposed by the original authors [25] nor the generally acknowledged three factor model results of prior CFA studies [e.g., 17, 18].

Further investigations into the common and specific variances captured by the MHC-SF total scale and the three subscales have revealed the overall strength of the GWB factor relative to the group factors. Like prior studies [e.g. 47, 48, 52], the GWB factor contributed to two-thirds or more of the total and common variance, whilst the contribution of the specific factors ranged from 3% to 24%. Notably, the PWB subscale captured lower contribution of unique variances when compared to its two counterparts (i.e., EWB and SWB subscale), suggesting that PWB is not as distinct of a standalone factor and that it largely overlaps with the entire scale. A possible explanation could be the broadness of the six theoretical dimensions of PWB proposed by Ryff [30]. For instance, the dimension of *positive relations with others* could be interpreted as part of the multidimensional model of SWB, while the dimension of *purpose in life* is highly related to the *interest* and *life satisfaction* dimensions found in EWB. However, due to the overlap, as suggested by Żemojtel-Piotrowska, Piotrowski [48], if a shorter instrument is needed, utilising the PWB subscale alone may be adequately effective.

More importantly, while the results of the study suggested that EWB and SWB factors do uniquely contribute to well-being, and there is a qualitative distinction between the contribution of the GWB factor and the three specific factors. The low MacDonald's $\omega_S$ coefficients and factor loadings strongly suggested that the three subscales do not yield precise and distinct enough measures of well-being to be utilised in research or clinical applications. Simultaneously, the high MacDonald's $\omega_H$ coefficients and ECV values of the GWB factor across both samples indicated that the MHC-SF ought to be considered as unidimensional. Thus, the calculation of separate subscale scores is not theoretically justified, and the current continuous scoring approach of an aggregated score across all 14 items ought to be the default. This also highlighted the limitations within the categorical diagnosis scoring system proposed by Keyes [34]. To be identified as either languishing or flourishing mental health, the individual must obtain specific cut-off requirements within the various group factors (i.e., EWB, SWB, and PWB). Consequently, due to the requirement of needing individualised interpretation of each subscale score, the categorical scoring system heavily contradicts the unidimensionality of the MHC-SF model, and ought to be avoided. It would be more appropriate to score the MHC-SF via the continuous scoring system and interpret the results using the total MHC-SF score as an indicator of the respondent's mental health, at least within the context of Singapore and Australia.

The unique trend of the common variance captured by the GWB factor, first reported by Żemojtel-Piotrowska, Piotrowski [48], was also identified in the current study. Across the results of 38 countries, Żemojtel-Piotrowska, Piotrowski [48] identified that in countries with

high collectivism such as Nepal, Japan, and India, the strength of the GWB factor tended to be stronger than individualistic countries such as Germany and United Kingdom. Within the current study, Singapore, a comparatively collectivistic country (Tan & Goh, 2006), yielded an ECV value of .09 higher than the Australia sample. While the ECV value of both countries reported in the current study (.85 for Singapore and .76 for Australia) are equal or higher than all the 38 countries reported (.40 to .76), the ECV value gap between Singapore (collectivistic) and Australia (individualistic) was observable in the current study.

Żemojtel-Piotrowska, Piotrowski [48] proposed two potential explanations for the differences in common variance captured by the GWB factor between collectivistic and individualistic countries. The current study concurs with the possibility of the first, but was able to dismiss the second. Firstly, Żemojtel-Piotrowska, Piotrowski [48] drawing upon the information provided by Cross, Bacon [77], posited that due to the interdependent self-construal found within collectivistic countries, social and psychological well-being are likely to become less distinct domains of well-being, further strengthening the contribution of the GWB factor. This is a potential explanation, seeing the limited contributions by the SWB and PWB factors found in the current study. However, that possibility would require further investigation, because the contribution of the SWB factor was found to be similar to that of the EWB factor within the current study. This result suggested that the interdependent self-construal found within collectivistic countries only affected the psychological domain of well-being and not the social domain.

Żemojtel-Piotrowska, Piotrowski [48], drawing upon the understanding put forth by Harzing [78], also offered a second plausible explanation, which explains that the differences in the strength of the GWB factor could be contributed to the stronger effects of acquiescence (i.e., compliance) found in collectivistic countries. They noted that as there were no reverse-scored items for the MHC-SF, individuals from collectivistic countries were hypothesised to be more likely to pick similar responses for all the questions, thus, contributing to the common factor variance. However, this explanation was not supported by the results of the current study. If it were to be true, the total MHC-SF score of the Singapore sample should see a significantly smaller variance compared to the Australia sample, however the total MHC-SF score variance reported by both samples were largely similar (Singapore [$SD$ = 1.30] and Australia [$SD$ = 1.40]).

With regard to the other psychometric properties of the MHC-SF, both samples were found to have excellent internal consistency. Across both samples, Cronbach's α was above .94 for the total scale and above .87 for the three subscales, while the MacDonald's ω reliability was above .96 for the GWB factor and above .90 for the three specific factors. Compared to prior studies (e.g., de Bruin & du Plessis, 2015; Keyes et al., 2008; Lamers et al., 2011; Żemojtel-Piotrowska et al., 2018), the current study reported equal or greater internal consistency reliability. These findings provided support that the items of MHC-SF, especially within the Singapore and Australia context, were reliably measuring the construct that they were designed to (i.e., scores of all the items on each of the subscale reported similar responses). However, the high Cronbach's α (.94) and MacDonald's ω (.96) of the total MHC-SF scale may not be necessarily desirable, as it may be indicative that some of the items are redundant [79]. Drawing back to the low factor loadings and insignificant contributions of unique variances by the PWB factor across both samples, it is highly possible that the undesired high reliability coefficients obtained and some of the redundant items may be from the PWB factor.

Concurrent validity of the MHC-SF was demonstrated via strong correlations ($r \geq$ .66) between the MHC-SF total scale and the three subscales with the WHO-5, which measures an individual's current mental well-being based on positive mood, vitality, and general interest. Notably, the correlations for the GWB, EWB, and PWB scale scores across both samples were

similar, while the correlations for the SWB scale were uniformly lower. This result is expected, as the items on the WHO-5 matches very closely to EWB followed by PWB, while no items were related to SWB.

To the best of our knowledge, criterion validity in the form of a singular self-reported question on state of mental health has not been investigated by prior studies. Results of the current study provided support to the criterion validity of the MHC-SF, as the MHC-SF total scale and the three subscales correlated moderately to strongly with a self-rated question on their state of mental health ($r \geq .52$) across both samples. In addition, the correlations observed were lower than that with the WHO-5, highlighting that a singular broad question was not able to fully capture the various dimensions of mental health and the ability of the MHC-SF to measure the nuances of mental health.

Discriminant validity of the MHC-SF was demonstrated by the moderate to high negative correlations between the MHC-SF total scale and the three subscales with the two subscales of the HADS ($r$ = -.45 to -.71) across both samples. The current results were higher than the previously reported correlations ($r$ = -.33 to -.56) by Doré et al. (2017). However, considering the predictiveness of mental health on mental illness and the intimate relationship between the two [20–22], results of the current study still demonstrated discriminant validity. In addition, it also provided support to the two-continua model; that mental health and mental illness are distinct but highly correlated constructs, and that symptoms of psychopathology do not imply the presence or absence of mental well-being.

### Limitations and further research

The current study only conducted the psychometric evaluation of the MHC-SF's structural, concurrent, criterion, and discriminant validity. Other forms of validity such as multigroup invariance (e.g., dividing the groups between students and workers or by gender), known-group validity (e.g., comparing stressed and non-stressed samples), or predictive validity (e.g., development of a mental disorder) could have been included to significantly add to the strength of the findings.

In addition, samples of the current study might have not been representative of the respective country's population. Due to the use of non-random sampling methods, convenience and snowball sampling, more than 70% of the participants were engaged with the study for course credit. In addition, the sampling venues (i.e., social media sites) may have further homogenised the results across the two countries despite the assumed collectivistic and individualistic differences. Hence, future evaluation studies of the MHC-SF should prioritise larger and more heterogenous samples, accounting for variables such as exposure to stressful events, socioeconomic, employment status, physical and mental health status, among others.

Including the current study, there have been numerous reports of the PWB factor contributing significantly lesser unique variances to the overall measure when compared to its counterparts, and its six items demonstrating weak factor loadings [e.g. 47, 48, 52]. Future research could investigate the theoretical dimensions of the PWB factor and explore whether they significantly overlap with the dimensions of the EWB and SWB factors, which could lead to the removing of redundant items or utilising the PWB subscale as a standalone instrument.

### Conclusion

Results of the current study are congruent with prior studies showing that the MHC-SF is a reliable and valid measure of mental health.

The results of the current study conclusively reported that a bifactor model of MHC-SF (a GWB factor and three specific factors of EWB, SWB, and PWB) demonstrated good to

excellent fit to the data, and was significantly better than any of the previously proposed model (i.e., one-, two-, and three-factor models).

Further investigations also identified that the EWB and SWB factors of MHC-SF contributed uniquely to the measure, while the PWB factor was similar to the GWB factor and had little to no unique contribution to the overall construct. Importantly, the results highlighted the strength of the GWB factor, which further lends its support to the existing aggregated scoring system of the MHC-SF utilised by researchers. Consequently, the use of the categorical diagnosis scoring system ought to be conducted with caution, as this approach requires interpretations of the subscale scores according to the hedonic or eudaimonic representation.

In sum, the current study provided strong supportive evidence that the MHC-SF demonstrates good psychometric properties within the Singapore and Australia context. Results also lend support to the unidimensionality of the MHC-SF and advocate for the utilisation of an aggregated scoring system over the categorical diagnosis. The MHC-SF appears to be a valid standardised measure, reliable, theory-driven, and comprehensive measure of mental health. Consequently, and because the MHC-SF has cross-cultural validity, the current study strongly encourages researchers, clinicians, and mental health professionals to start utilising the instrument and promoting awareness of mental health.

## Author Contributions

**Conceptualization:** Zack Zhishen Yeo, Lidia Suárez.

**Data curation:** Zack Zhishen Yeo.

**Formal analysis:** Zack Zhishen Yeo, Lidia Suárez.

**Investigation:** Zack Zhishen Yeo.

**Methodology:** Zack Zhishen Yeo.

**Project administration:** Zack Zhishen Yeo.

**Resources:** Zack Zhishen Yeo.

**Software:** Zack Zhishen Yeo.

**Supervision:** Lidia Suárez.

**Validation:** Zack Zhishen Yeo, Lidia Suárez.

**Visualization:** Zack Zhishen Yeo.

**Writing – original draft:** Zack Zhishen Yeo.

**Writing – review & editing:** Zack Zhishen Yeo, Lidia Suárez.

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
