## [Decision Letter · Decision Letter 0]

10 Jan 2022

PONE-D-21-16159Validation of the Mental Health Continuum-Short Form: The Bifactor Model of Emotional, Social, and Psychological Well-beingPLOS ONE

Dear Dr. Yeo,

Thank you for submitting your manuscript to PLOS ONE. After careful consideration, we feel that it has merit but does not fully meet PLOS ONE’s publication criteria as it currently stands. Therefore, we invite you to submit a revised version of the manuscript that addresses the points raised during the review process.

We look forward to receiving your revised manuscript.

Kind regards,

Francesca Chiesi

Academic Editor

PLOS ONE

Journal Requirements:

3. Please upload a new copy of Figure 1 as the detail is not clear. Please follow the link for more information: https://blogs.plos.org/plos/2019/06/looking-good-tips-for-creating-your-plos-figures-graphics/" https://blogs.plos.org/plos/2019/06/looking-good-tips-for-creating-your-plos-figures-graphics/

Reviewers' comments:

Reviewer's Responses to Questions

**Comments to the Author**

1. Is the manuscript technically sound, and do the data support the conclusions?

Reviewer #1: Partly

Reviewer #2: Yes

2. Has the statistical analysis been performed appropriately and rigorously? 

Reviewer #1: Yes

Reviewer #2: Yes

3. Have the authors made all data underlying the findings in their manuscript fully available?

Reviewer #1: No

Reviewer #2: Yes

4. Is the manuscript presented in an intelligible fashion and written in standard English?

Reviewer #1: No

Reviewer #2: Yes

5. Review Comments to the Author

Reviewer #1: Psychometric evaluations of instruments measures of wellbeing are important since their use in epidemiological surveillance and also various wellbeing interventions are increasing rapidly.

For a standard evaluation, of an already existing well-known instrument, the submitted paper is very long, over 47 pages with references. Basically all sections of the manuscript could be shortened, some quite drastically so - starting with the background text leading up to the aims and hypothesis on page 14. The relevant text for the psychometric evaluation starts at line 201. If the authors want to give a general background to the field it should be no longer than a couple of paragraphs, not 5 pages. Concentrate the background (2-3 pages) on what has been done thus far in terms of psychometric evaluations, identify and make it clear to the reader what the problems have been previously with these evaluations and how your study contributes to solving those problems or add knowledge.

Reviewer #2: The authors have clearly laid out the literature review for this research topic, in a succinct yet comprehensive manner. I am especially impressed by the coherence and seamless transitions of the introduction and literature review parts. The authors pointed out the shortcomings of other measures of wellbeing in the literature to justify the validation of the measure in this study.

The Mental Health Continuum Short Form (MHC-SF) has been adequately described before delving into the methodology, including critical remarks of the scale. The authors provided a fair presentation of the pros and cons of the tri-factor vs bi-factor models of MHC-SF, and established strong grounds for justifying their choice of the bi-factor model. The research hypotheses were concise and well-worded.

The literature review referred to mostly global studies. Considering this study is validated in Singapore and Australia, it is recommended to include more specific references about previously validated wellbeing scales in these countries, and it is recommended to describe the local need for such scales.

I had concerns about the snowball sampling technique, but it was fairly described as a limitation at the end and the impact on the generalizability of the results was outlined. However, it should noted that the sampling venues (social media sites) may have also played a role in homogenizing the results across two counties despite the collectivist/individualist differences.

Every decision and step in the analysis was explicitly justified. The interpretation of the results was comprehensive and provided sound reasoning to support a unidimensional structure.

The only point of concern was the sampling method used, but overall this is well written paper with a robust methodological basis. It significantly adds to the literature of MHC-SF and provides international practitioners with another valid instrument to implement. I strongly recommend this paper for publication.

6. PLOS authors have the option to publish the peer review history of their article (what does this mean?). If published, this will include your full peer review and any attached files.

Reviewer #1: No

Reviewer #2: **Yes: **Saad Yaaqeib

---

## [Author Response · Author response to Decision Letter 0]

7 Mar 2022

Dear Reviewers, 

Thank you for taking your time to review and comment on the paper titled,” Validation of the mental health continuum-short form: The bifactor model of emotional, social, and psychological well-being”

The followings are the changes that we have made to the paper in accordance with your feedback:

We have acknowledged Reviewer #1’s feedback regarding the lengthy introduction and background information given. We have significantly reduced the background information and have limited the introduction of each core concept to only one paragraph. However, we have also recognised the importance of highlighting the shortcomings of other measures, the adequate description, and prior psychometric evaluations of the current measure, as per commented by Reviewer #2, hence we have kept majority of the latter part of the introduction. We hope that by doing so, we provide readers a succinct yet comprehensive literature review of the matter. 

Reviewer’s #2 recommendation of including more specific references to previously validated wellbeing scales within local context was not able to be addressed because we followed Reviewer #1’s recommendations to focus on the scale and it’s psychometrics. With regard to highlighting the local need of the scale, we hope by extensively highlighting the flaws of existing measures and having noted the lack of validation studies of the tool in local context, we can justify the need for the current investigation into the measure. 

In addition, we have also recognised the concern raised by Reviewer #2 regarding the sampling method, and thus, have added more comments within the limitations section to highlight the potential generalisation issues. 

The data of the study have been made public and can be found on figshare’s depository. https://doi.org/10.6084/m9.figshare.19174205.v2.

Lastly, we have made changes to the manuscript so it now meets PLOS ONE’s style requirements, as requested by the Academic Editor.

Once again, thank you for your time reviewing and providing valuable comments on the paper. I hope we have adequately addressed the concerns that each of you have raised.

Best Regards, 

Zack Yeo Zhishen

Lidia Suarez

---

## [Editor Report · Decision Letter 1]

22 Mar 2022

PONE-D-21-16159R1Validation of the mental health continuum-short form: The bifactor model of emotional, social, and psychological well-beingPLOS ONE

Dear Dr. Yeo,

Thank you for submitting your manuscript to PLOS ONE. After careful consideration, we feel that it has merit but does not fully meet PLOS ONE’s publication criteria as it currently stands. Therefore, we invite you to submit a revised version of the manuscript that addresses the points raised during the review process.

The points raised by the reviewers were adequately addressed and the paper is significantly improved. I just suggest completing the well-being measures review reported in the Introduction with the following:

https://journals.plos.org/plosone/article?id=10.1371/journal.pone.0252709

We look forward to receiving your revised manuscript.

Kind regards,

Francesca Chiesi

Academic Editor

PLOS ONE
---

## [Author Response · Author response to Decision Letter 1]

24 Apr 2022

Dear Reviewers, 

Thank you for taking your time to review and comment on the paper titled,” Validation of the mental health continuum-short form: The bifactor model of emotional, social, and psychological well-being”

The followings are the changes that we have made to the paper in accordance with your feedback:

We have acknowledged your feedback of completing the well-being measures review section of the paper by including the recent well-being instrument developed and published by Bonacchi et al. (2021). We have added a new paragraph towards the end of the section (line 147-152) to acknowledge the instrument, including its pros and limitations. 

Once again, thank you for your time reviewing and providing valuable comments on the paper. I hope we have adequately addressed the feedback.

Best Regards, 

Zack Yeo Zhishen

Lidia Suarez

---

## [Editor Report · Decision Letter 2]

26 Apr 2022

Validation of the mental health continuum-short form: The bifactor model of emotional, social, and psychological well-being

PONE-D-21-16159R2

Dear Dr. Yeo,

We’re pleased to inform you that your manuscript has been judged scientifically suitable for publication and will be formally accepted for publication once it meets all outstanding technical requirements.

Kind regards,

Francesca Chiesi

Academic Editor

PLOS ONE
---

## [Editor Report · Acceptance letter]

29 Apr 2022

PONE-D-21-16159R2 

Validation of the mental health continuum-short form: The bifactor model of emotional, social, and psychological well-being 

Dear Dr. Yeo:

I'm pleased to inform you that your manuscript has been deemed suitable for publication in PLOS ONE. Congratulations! Your manuscript is now with our production department. 

Kind regards, 

on behalf of

Dr. Francesca Chiesi 

Academic Editor

PLOS ONE